# Comparison of Kinetics of Antibody Avidity and IgG Subclasses’ Response in Patients with COVID-19 and Healthy Individuals Vaccinated with the BNT162B2 (Comirnaty, Pfizer/BioNTech) mRNA Vaccine

**DOI:** 10.3390/v15040970

**Published:** 2023-04-14

**Authors:** Waldemar Rastawicki, Rafał Gierczyński, Aleksandra Anna Zasada

**Affiliations:** 1Department of Bacteriology and Biocontamination Control, National Institute of Public Health NIH—National Research Institute, 00-791 Warsaw, Poland; 2Department of Sera and Vaccines Evaluation, National Institute of Public Health NIH—National Research Institute, 00-791 Warsaw, Poland

**Keywords:** antibody, COVID-19 vaccine, SARS-CoV-2, vaccination

## Abstract

There are limited reports concerning the levels of antibodies in IgG subclasses and the avidity of IgG, which is the functional strength with which an antibody binds to an antigen in serum samples obtained at different times after infection or vaccination. This study investigated the kinetics of antibody avidity and the IgG antibody response within IgG1-IgG4 subclasses in individuals vaccinated with the BNT162B2 mRNA vaccine and in COVID-19 patients. Serum samples were collected from individuals vaccinated with three doses of the BNT162B2 (Comirnaty, Pfizer/BioNTech) vaccine and from unvaccinated COVID-19 patients. This study revealed that IgG1 was a dominating subclass of IgG both in COVID-19 patients and in vaccinated individuals. The level of IgG4 and IgG avidity significantly increased 7 months after the first two doses of the vaccine and then again after the third dose. IgG2 and IgG3 levels were low in most individuals. Investigating IgG avidity and the dynamics of IgG subclasses is essential for understanding the mechanisms of protection against viral infections, including COVID-19, especially in the context of immunization with innovative mRNA vaccines and the possible future development and application of mRNA technology.

## 1. Introduction

There are many reports on the kinetics between the IgG, IgA, and IgM antibodies and the S1 and N proteins of the SARS-CoV-2 virus, both in patients infected with COVID-19 and healthy vaccinated individuals. However, there are only limited reports concerning the levels of antibodies in particular IgG subclasses and the avidity of IgG, which is the functional strength with which an antibody binds to an antigen in serum samples obtained at different times after infection or vaccination [1,2]. This information constitutes an important qualitative characteristic of immunity after infections of SARS-CoV-2, and could provide insights into the efficacy of anti-COVID-19 immunization.

Our previous study compared the IgG and IgA antibody responses to SARS-CoV-2 7 months after complete vaccination with two doses and after the third dose of the BNT162b2 vaccine in healthy adults [3]. In this study, the research is extended to include an investigation of the kinetics of antibody avidity and IgG antibody response in IgG1-IgG4 subclasses in a group of vaccinated individuals and, additionally, in patients with COVID-19.

## 2. Materials and Methods

The study population consisted of 47 hospitalized patients (25 women and 22 men; average age of 67.5 years) with COVID-19, which was confirmed by PCR, and 44 healthy subjects (28 women and 16 men; average age of 50.5 years) vaccinated three times with the BioNTech/Pfizer messenger RNA (mRNA) vaccine (COMIRNATY^®^/BNT162b2). All individuals signed a written consent form to participate in the study. The median time between the onset of symptoms and the first sample was 8.3 days. Second serum samples were taken about one week later for all of the patients with COVID-19. The first series of serum samples from vaccinated individuals was obtained an average of 25 days after receiving the second dose of the vaccine, with the second series obtained between 198 and 231 days after receiving the second dose. The third series included serum samples collected an average of 30 days after the third dose of the same vaccine (7 months after the second dose). Four individuals had a history of COVID-19 prior to their first dose of the vaccine. Additionally, seven serum samples were taken from a 56-year-old male at different times after acute COVID-19 in April 2020 and after vaccination with two doses of the BNT162b2 vaccine in January 2021.

Samples were tested using an in-house ELISA with recombinant S1 and N proteins (NativeAntigen, UK). The avidity index (AI) of IgG antibodies was determined as the ratio of the optical density (OD_450_) obtained in a reaction with a dissociating solution (1.0 M of ammonium thiocyanate) to a value obtained with a control solution. The avidity index (AI) was interpreted arbitrarily according to the range: <40 AI (low avidity), 40–60 AI (medium avidity), and >60 AI (high avidity). The level of antibodies in the individual subclasses (IgG1, IgG2, IgG3, and IgG4) was determined by the absorbance value of light at a wavelength of 450 nm (OD_450_). To allow assay standardization and to monitor quality control, including intra- and inter-plate variability, we used serum samples with known recent infection and well-characterized antibody status. The specificity of the in-house ELISA was tested using serum samples obtained from individuals with various bacterial and viral infections. Positive and negative controls were used in each study series.

The significance of differences in the frequency of detection of antibodies depending on sex and age was assessed by a chi-square test of independence using Yates’s correction. The differences were considered statistically significant when the *p*-value significance levels were lower than α = 0.05.

## 3. Results

The results of the study showed a significant increase in the avidity and level of the IgG1 subclass antibodies for the S1 and N proteins of SARS-CoV-2 in serum samples obtained twice from patients with COVID-19. The IgG2-IgG4 subclass antibodies for the S1 protein were not detected or were detected at a low level in patients with an acute SARS-CoV-2 infection. Relatively high levels of the IgG2, IgG3, and IgG4 antibodies (OD_450_ > 1.0) for the N protein were detected only in 2, 11, and 6 patients, respectively.

All four vaccinated individuals with a history of COVID-19 exhibited a high IgG antibody avidity for both proteins in the first-tested serum samples. However, while the avidity of antibodies for the N protein did not change, the avidity of antibodies for the S1 protein increased 6 months later, reaching a maximum level of 93.1% after the third dose of the vaccine. The study also showed very high levels of the IgG1 antibodies and low levels of the IgG4 antibodies for the S1 protein in serum samples obtained from individuals with a history of COVID-19 after the first two doses of the BNT162b2 vaccine. After 6 more months, only a slight decrease in the geometric mean (GM) of the IgG1 level and a considerable increase in the GM of the IgG4 level was detected. The IgG1 and IgG4 antibodies reached their maximum level after the booster. In vaccinated individuals with a history of COVID-19, the level of antibodies for the nucleocapsid was lower than the level of antibodies for the S1 protein and remained similar in all three series of the study. 

Vaccinated individuals without a history of COVID-19 were only tested for antibodies for the S1 protein. The avidity of IgG antibodies in samples obtained in the first series of the study was lower than in vaccinated individuals with a history of COVID-19. The avidity of IgG antibodies was higher in serum collected in the second series of the study and reached the maximum (GM 96.3%) after the third dose of the BNT162b2 vaccine. Similarly, the average level of IgG1 antibodies in vaccinated individuals without a history of the illness was lower than in vaccinated individuals with a history of COVID-19. Furthermore, a faster decrease in the level of the IgG1 antibodies was observed in the samples obtained 6 months later. As in the previous group, the IgG4 subclass antibodies reached a maximum value after the third dose of the vaccine (Table 1). The IgG2 antibodies for the S1 protein were not detected or were present at very low levels in vaccinated individuals. Relatively high levels of the IgG3 antibodies (OD_450_ >1.0) for the S1 protein were detected in only eight cases (18%). 

In the first serum sample obtained from a 56-year-old male 2 months after infection with COVID-19, antibodies for the S1 and N proteins were detected (Figure 1). Consecutive serum samples showed a consistent decrease in the IgG1 and increase in the IgG4 subclass antibodies for the N protein. The IgG1 antibodies for the S1 protein remained at similar levels for several months and, subsequently, increased rapidly after two doses of the BNT162b2 vaccine. Interestingly, this patient had a low level of the IgG4 subclass antibodies for the S1 protein as well as a low level of the IgG2 and IgG3 antibodies for the S1 and N proteins, both after the illness and after two doses of the vaccine. The IgG antibodies for both proteins were characterized by high avidity. After vaccination, an increase in avidity was observed only for antibodies specific to the S1 protein. 

A statistical analysis of the results showed that high-avidity (>60 AI) IgG antibodies and high-level (OD_450_ ≥ 2.5) IgG1 and IgG4 antibodies were detected in individuals below 58 years old more often than in older persons (chi^2^ = 20.47, 18.77, and 11.88, respectively; *p* < 0.05). No statistical differences were observed depending on the sex of the individuals (*p* > 0.05).

## 4. Discussion

Investigation of the avidity of IgG antibodies as well as specific antibodies in individual IgG subclasses offers a better insight into the kinetics of the humoral response over the course of the COVID-19 infection, allowing us to assess the effectiveness of preventive vaccinations [4,5,6].

In general, viral infections lead to an increase in the levels of IgG antibodies of the IgG1 and IgG3 subclasses, with IgG3 antibodies appearing first during the course of the infection. However, IgG3-dominated responses appear to be rare. This is caused by the fact that IgG1 is the most abundant immunoglobulin in human serum, making up about 65% of the total amount of IgG, while IgG3 accounts only for 7% [7,8]. Our study confirmed the IgG1 dominance, but a high level of IgG3 for the SARS-CoV-2 virus was observed in a few individuals. However, in the study of Korobova et al. [9] IgG3 showed the most prominent concentration dynamics in COVID-19 patients. According to the authors, this may be due to the specific structure or function of IgG3 [9].

Fluctuations in subclass persistence can be observed during the course of COVID-19. IgG3 levels decreased, whereas IgG1 and IgG2 levels were seen to rise over time [9]. In this study, we only tested samples obtained twice during the initial phase of the disease. Nevertheless, our data revealed an increase in IgG1 and IgG3 levels after a week of illness, especially for the N protein, while IgG2 and IgG4 levels remained less detectable during the study period. Similar results were also obtained by others [10,11].

When examining the humoral response after vaccination, much more interesting results were obtained about the fluctuations in subclass persistence. Seven months after the two doses of the vaccine, there was a decrease in IgG1 and a very clear increase in IgG4 antibodies. Irrgang et al. [6] and Poolchanuan et al. [5] also observed a significant increase in IgG4 levels after vaccination with the second dose of the mRNA vaccine, but not after vaccination with adenoviral vectors or an inactivated vaccine. IgG4 levels were further boosted by the third dose of the mRNA vaccine. A good example of the different kinetics of the IgG1 and IgG4 antibodies is also the humoral response to the N protein in a 56-year-old male with a history of COVID-19. Differences in the kinetics of the IgG1 and IgG4 subclass antibodies result from their specific structure and function. The characteristic feature in the immune regulation of IgG4 is its tendency to appear only after prolonged immunization [12]. It usually takes many months before IgG4 responses become prominent. It is worth mentioning that, so far, only a few studies on the role of vaccine-induced IgG4 responses against COVID-19 are available. Our results, similar to results obtained by Irrgang et al. [6], clearly demonstrate that IgG4 is becoming the most dominant among all IgG subclasses in some vaccinated individuals, especially several months after immunization. These are important observations, and the mechanisms should be investigated deeply because mRNA technology has huge future potential to be used not only for the development of vaccines against other pathogens, but also as a therapy for non-infectious diseases such as cancer. 

Avidity maturation is crucial for protection against viral infections [13]. IgG avidity grows after SARS-CoV-2 infection and after vaccination [1,13]. However, Bauer et al. [13] noticed that the avidity of IgG towards SARS-CoV-2 antigens in COVID-19 patients is usually low due to incomplete avidity maturation. Incomplete avidity maturation is related to a discontinuous kinetics of avidity maturation unrelated to the time of observation. Vaccination might be more effective than infection because all vaccinated individuals had a high avidity index of anti-S1 IgG in the study, whereas only 12.8% and 46.8% of unvaccinated COVID-19 patients had a high avidity index of anti-S1 and anti-N, respectively. The avidity increased over time and with successive doses of the vaccine. It seems that the prolonged availability of antigens is required for proper avidity maturation. Nakagama et al. [2] revealed that high-avidity antibodies might provide better long-lasting protection against evolving variants of SARS-CoV-2. 

Investigating IgG avidity and the dynamics of IgG subclasses is essential for understanding the mechanisms of protection against viral infections, including COVID-19, especially in the context of immunization with innovative mRNA vaccines and the possible future development and application of mRNA technology. 

## Figures and Tables

**Figure 1 viruses-15-00970-f001:**
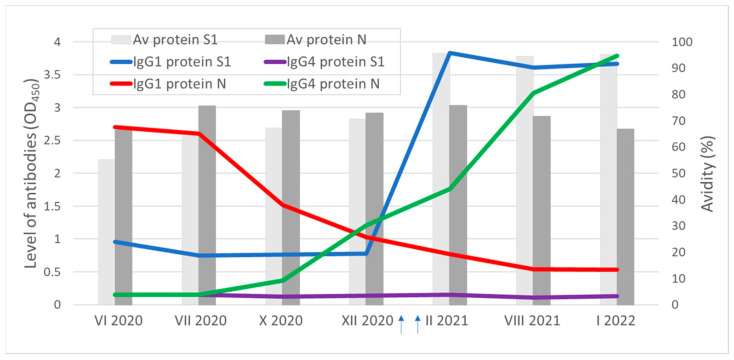
Dynamics of avidity and the level of IgG1 and IgG4 subclass antibodies (OD_450_) for the S1 and N proteins of SARS-CoV-2 in the seven serum samples obtained from the 56-year-old male at different times after COVID-19 infection in April 2020 and after vaccination. Arrows show the time of vaccination with the two doses of the BNT162b2 vaccine.

**Table 1 viruses-15-00970-t001:** The avidity of the IgG antibodies and levels of IgG1-IgG4 subclass antibodies for the S1 and N protein of the SARS-CoV-2 virus in unvaccinated patients with COVID-19, vaccinated individuals with a history of COVID-19 and vaccinated individuals without a history of COVID-19 in relation to the test series.

Group and Number of Tested Individuals	Test Series	Antigen	Avidity Index of IgG (Number, Percentage of Individuals, and Geometric Mean Value)				GM of the Levels of Antibodies (OD_450_) in the Subclasses			
			<40 AI	40–60 AI	>60 AI	GM	IgG1	IgG2	IgG3	IgG4
unvaccinated patients with COVID-19 (no. 47)	1	S1	44 (93.6%)	3 (6.4%)	-	28.4%	0.41	0.11	0.10	0.18
	2		6 (12.8%)	35 (74.5%)	6 (12.8%)	46.7%	0.95	0.10	0.11	0.20
	1	N	16 (34.0%)	29 (61.7%)	2 (4.3%)	43.7%	0.91	0.11	0.15	0.18
	2		4 (8.5%)	21 (44.7%)	22 (46.8%)	59.6%	2.20	0.11	0.40	0.24
vaccinated individuals with a history of COVID-19 (no. 4)	1	S1	-	-	4 (100%)	87.3%	3.71	0.08	0.07	0.27
	2		-	-	4 (100%)	89.1%	2.28	0.07	0.05	1.13
	3		-	-	4 (100%)	93.1%	3.30	0.08	0.07	1.91
	1	N	-	-	4 (100%)	65.1%	0.70	0.08	0.20	0.30
	2		-	-	4 (100%)	66.5%	0.35	0.08	0.16	0.28
	3		-	-	4 (100%)	66.9%	0.39	0.07	0.18	0.31
vaccinated individuals without a history of COVID-19 (no. 40)	1	S1	2 (5.0%)	18 (45.0%)	20 (50.0%)	62.4%	2.14	0.08	0.22	0.26
	2		-	4 (10.0%)	36 (90.0%)	77.5%	0.81	0.10	0.14	1.94
	3		-	-	40 (100%)	96.3%	2.85	0.21	0.20	3.50

## Data Availability

No new data were created or analyzed in this study. Data sharing is not applicable to this article.

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
