# Peer review of "Comparison of Kinetics of Antibody Avidity and IgG Subclasses’ Response in Patients with COVID-19 and Healthy Individuals Vaccinated with the BNT162B2 (Comirnaty, Pfizer/BioNTech) mRNA Vaccine"

_viruses, 2023, doi:10.3390/v15040970_

Round 1

Reviewer 1 Report

The Authors present the comparison of kinetics of antibody avidity and IgG subclasses response in patients with COVID-19 and vaccinated healthy individuals. The analysis comprised 47 patients with COVID-19 and 44 healthy controls, including 4 with past infection. The analysis is a continuation of a previously published study (item 3 of 7 papers cited). The study is a continuation of a previously published data (item 3 of 7 papers cited). It was designed correctly, the biggest advantage being the evaluation including IgG subclasses. However, the discussion should be deepened and include other works evaluating the kinetics of changes in IgG subclasses, e.g. PMID: 35632683, PMID: 34480056. Factors that affect the kinetics of IgG subclasses should be discussed. The section 0. How to use this template should be removed.

Author Response

Dear Reviewer,

Thank you for the very helpful suggestions. The discussion has been deepened. Six other works about the kinetics of IgG subclasses have been included. The section 0 has been removed.

Reviewer 2 Report

The article compares data on immunoglobulin subclasses between patients with COVID-19 and vaccinated individuals. The data are clear and well presented, the conclusions supported by the results.

However, a remark can be made on the methodology used. Vaccinated individuals without a history of COVID-19 were tested against the S1 protein only. However, given the high circulation of the virus in the population, asymptomatic infection cannot be excluded for these individuals. The search for antibodies against the N protein could clarify whether these individuals actually did not suffer an asymptomatic COVID-19 infection

Author Response

Dear Reviewer,               

We agree that some of the vaccinated individuals might had asymptomatic infection. However, the aim of the study was to analyze and compare the humoral response in two clearly defined groups of persons: those with symptomatic disease and those who were vaccinated. Moreover, our previous study, as well as study conducted by other researchers, revealed that the humoral response is much weaker in asymptomatic individuals than in fully symptomatic individuals. Nevertheless, we plan to continue the study the humoral response after the fourth dose of the vaccine to answer the questions: how long the post-vaccination antibodies to S1 protein persist and how many individuals had COVID-19 infection (based on the level of anty-N antibodies) despite vaccination with four doses of the vaccine.

Reviewer 3 Report

In the manuscript, Waldemar Rastawicki and colleagues performed research on the

 kinetics of antibody avidity and IgG subclass response in patients with COVID-19 and healthy individuals vaccinated with the BNT162b2 (Comirnaty, Pfizer/BioNTech)  mRNA vaccine. The study provided an essential for understanding the the dynamics of IgG avidity and IgG subclasses. However, some issues need to be addressed to strengthen the conclusions and the manuscript.

1. In Page1 line 26-32, please delete the irrelevant section”0. How to Use This Template” in the manuscript.

2. In Page2 line54,what does COVID-12 mean?

3. In the manuscript, the author using an in-house ELISA with recombinant S1 and N proteins for detecting antibody avidity, IgG1, IgG2, IgG3 and IgG4, Whether there are some corresponding measures for the quality control of in-house ELISA, the low level OD450 values of IgG2 and IgG3, which is due to methodological problems or low levels of the samples themselves need to confirm.

4. In addition, the sample size of the vaccinated individuals after past COVID-19 is low(only 4).

Author Response

Dear Reviewer,               

Thank you for the very helpful suggestions. Below are our point by point responses:

  1. In Page1 line 26-32, please delete the irrelevant section”0. How to Use This Template” in the manuscript. - delated
  2. In Page2 line54,what does COVID-12 mean? – corrected to COVID-19
  3. In the manuscript, the author using an in-house ELISA with recombinant S1 and N proteins for detecting antibody avidity, IgG1, IgG2, IgG3 and IgG4, Whether there are some corresponding measures for the quality control of in-house ELISA, the low level OD450 values of IgG2 and IgG3, which is due to methodological problems or low levels of the samples themselves need to confirm. – the information concerning standardization of the assay has been added in the part 2. Materials and Methods
  4. In addition, the sample size of the vaccinated individuals after past COVID-19 is low(only 4). – we agree that the sample size is very low, but we decided to separate this group from the group of vaccinated individuals for greater precision of the study

Round 2

Reviewer 1 Report

No comments.

Reviewer 2 Report

No further comments. Methods should be improved in future studies

Reviewer 3 Report

The authors have provided a full explanation addressing all of the concerns that came to my attention.